# Radiofrequency Thermal Ablation for the Treatment of Chronic Insufficiency of the Saphenous Vein—A Comparative Retrospective Study

**DOI:** 10.3390/ijerph20043308

**Published:** 2023-02-14

**Authors:** Octavian Andercou, Bogdan Stancu, Horațiu Flaviu Coman, Beatrix Cucuruz, Thomas Noppeney, Dorin Marian

**Affiliations:** 1Department of Surgery, University of Medicine and Pharmacy “Iuliu Hatieganu”, 400347 Cluj-Napoca, Romania; 2Department of Vascular Surgery, County Emergency Hospital, 400347 Cluj-Napoca, Romania; 3Department of Vascular Surgery, Martha Maria Hospital Nuremberg, 90491 Nuremberg, Germany; 4Department of Vascular and Endovascular Surgery, University Hospital Regensburg, 93053 Regensburg, Germany; 5Department of Surgery, University of Medicine and Pharmacy “George Palade”, 540142 Targu Mures, Romania

**Keywords:** varicose veins, radiofrequency ablation, surgical treatment

## Abstract

Objectives: The broad spectrum of chronic venous disease encompasses varicose veins, edema, hyperpigmentation and venous ulcers. Radiofrequency thermal ablation is indicated for the treatment of superficial venous reflux of the lower limb. Our research is a comparative clinical study that aims to identify the most effective and safest therapeutic method in the management of chronic venous insufficiency of the lower limbs. Materials and methods: Patients admitted to the Department of Surgery of the University of Medicine and Pharmacy in Cluj-Napoca, Romania, with the clinical diagnosis of varicose veins of the lower limbs, treated by thermal ablation with radiofrequency or by open surgical techniques during the year 2022, were included. Results: A percentage of 50.9% of the patients were treated by the radiofrequency thermal ablation procedure and 49.1% by surgical treatment. More than half of them were hospitalized for 2 days. The duration of hospitalization was significantly longer in patients who had postoperative complications (*p* < 0.001). The chance of being treated by open surgical treatment for a small saphenous vein is 10.11 times higher than by radiofrequency thermal ablation. Conclusion: According to the applied tests, there is no statistical difference between the group treated by radiofrequency thermal ablation and the one surgically treated in terms of sex, age, origin, CEAP clinical stage at hospitalization, clinical diagnosis at hospitalization and affected lower limb.

## 1. Introduction

Chronic venous insufficiency is an extremely common pathology, with an increasing incidence in developed countries, which conveys a major impact on both individual health and the health system.

The prevalence of chronic venous disease in industrialized countries amounts to 30–60% of the population. The annual incidence of varicose veins is estimated at 2.6% for females and 1.9% for males, according to the Framingham study [1]. The main risk factors for varicose veins are advanced age, female gender, family history of varicose veins, prolonged orthostatism and sedentarism [2,3].

Current interventional treatment options include high ligation of the saphenous vein, ligation and stripping, ambulatory phlebectomy, endovenous thermal ablation and non-thermal options. Until the introduction of the new techniques, great saphenous vein (GSV) high-ligation and stripping was the reference surgical technique in the curative treatment of varicose veins. During preoperative marking for the great saphenous vein (GSV) ligation and stripping procedure, the surgeon should highlight the extent and distribution of venous reflux in the lower limb. High ligation of the GSV is performed close to the femoral vein; however, it is down with caution so as not to narrow the lumen of the femoral vein. Nowadays, minimally invasive methods are gaining increasingly in popularity. Radiofrequency thermal ablation comprises a minimally invasive technique for efficiently treating venous reflux with minimal discomfort to the patient. One of the main advantages of the technique is that it can be performed on an outpatient basis using local anesthesia. Surgical treatment of varicose veins of the lower limbs will be individualized for each patient based on the preoperative evaluation and on the preference of the surgeon. According to this assessment, the extent of varicose veins in the great saphenous vein (GSV), small saphenous vein (SSV), and collateral and perforating veins will dictate how the various techniques of ligation, stripping and phlebectomy are combined.

According to the Clinical Practice Guidelines of the European Society for Vascular Surgery from 2022 [4], in the case of GSV reflux, first-line endovenous thermal ablation is recommended (recommendation level I A). In the case of SSV, the recommendations are similar (recommendation level I B). Concomitant phlebectomies are recommended to either be treated in the same session as the main trunk or in a secondary intervention (recommendation level II B) [4,5,6,7].

The present clinical study wishes to evaluate the outcomes of these recommendations if applied to the population of Romania, which has certain peculiarities compared to other European countries. A study conducted in 2018 by Feodor et al. [8] identified that the rate of diagnosis of chronic venous insufficiency is currently on a continuous increase, the rate of medical addressability is still low and stagnant in recent years, and the rate of patients presenting in a severe clinical stage is still high.

Regarding the therapeutic methods used for the treatment of varicose veins of the lower limbs, the present study will focus on two different therapeutic approaches, the most frequently used in the Department of Surgery of the University of Medicine and Pharmacy Cluj Napoca, Romania: open surgical treatment of GSV and SSV and radiofrequency thermal ablation (RFA). The RFA technique was performed by a single surgeon, whereas the open technique was by several surgeons. The aim of our study is to evaluate which of the two therapeutic methods has brought a superior benefit to the patient in terms of efficiency, safety and costs.

## 2. Materials and Methods

The inclusion criteria in the study were:Patients of the Surgery Clinic II from Cluj-Napoca between January and December 2022;Clinical diagnosis of varicose veins of the lower limbs;CEAP clinical classification of chronic venous insufficiency between C2 and C6;Treatment of varicose veins by radiofrequency thermal ablation or by open surgical treatment.

The exclusion criteria of the study were:CEAP clinical stage of C0 or C1 chronic venous insufficiency;Treatment of varicose veins by a therapeutic method other than the two stated in the inclusion criteria (conservative, intravenous LASER ablation, sclerotherapy).

The patient sample was divided into two distinct groups according to the procedure used: group I—surgical treatment (GSV high ligation, stripping and phlebectomies); group II—radiofrequency thermal ablation (for both internal and external saphenous veins).

In all patients, the reflux in either the GSV or the SSV was documented by duplex ultrasound (DUS). Additionally, it excluded obstruction in the deep veins.

The two types of therapeutic procedures targeted venous reflux of the GSV or SSV level, and both were accompanied by secondary procedures of phlebectomies to remove the reflux from the superficial varicose veins remaining after the main procedure.

### Surgical Technique

For radiofrequency ablation of the saphenous veins, we used the ClosureFast radiofrequency generator (Medtronic) with a 7 Fr 100 cm long ClosureFast catheter. The inserted catheter has an active tip with a length of 7 cm, which heats a 7 cm vein segment in one 20-s interval to shrink and collapse the target veins. The temperature is kept at a stable 120 degrees Celsius during the 20-s treatment cycle. The catheter delivers the energy which causes collagen contraction, which finally leads to obliteration of the lumen through endothelial destruction, inflammatory response, fibrosis and permanent occlusion.

The patient is placed in the reverse Trendelenburg position in order to obtain proper vein distension. Ultrasound-guided access to the great saphenous vein is obtained using a micropuncture needle and a 0.018-inch guide wire at the level of the distal thigh or just below the knee. A 7 Fr sheath is placed over the wire; the ClosureFast™ (VNUS Medical Technologies, INC, San Jose, CA, USA) catheter is then placed through the 7 Fr sheath and advanced to the level of the SFJ. The distance between the SFJ and the tip of the catheter should be between 15–20 mm.

The entire treatment length of the vein must be circumferentially injected with a normal saline solution prior to beginning ablation. The superficial fascia anterior to the vein and the muscular fascia posterior to the vein creates the saphenous canal. By using ultrasound guidance, the tumescent solution of about 300 to 450 mL of 0.9% saline is injected circumferentially around the vein in the saphenous canal, which is between the superficial fascia anterior to the vein and the posterior fascia to the vein, which separates it from the muscle.

RFA may begin after the appropriate instillation of tumescent solution along the entire treatment length of the vein, and the final positioning of the RF catheter is rechecked and confirmed to be 15–20 mm from the saphenofemoral junction (SFJ). Next, the catheter is retracted into 6.5 cm segments, and each vein segment is treated with a single 20-s cycle.

After the final segment of the vein has been treated, the catheter and the sheath are removed, and manual pressure is applied at the sheath insertion to achieve hemostasis. An elastic stocking or bandage is applied for 24 h postoperatively.

For the SSV RF ablation, the patient is placed in a ventral decubitus. In this case, we used a 60 cm long catheter, inserted just above the lateral malleolus and positioned close to the sapheno-popliteal junction. The following steps are the same as for the GSV.

In both techniques, open surgical and minimally invasive procedures, we performed simultaneously mini phlebectomies for the dilated veins.

Data acquired were analyzed according to demographics, pathology-related information and therapeutic procedure-related information.

The entire database was processed in order to obtain a representative sample for the entire population with this pathology, treated by one of the two therapeutic methods evaluated.

To meet the objective of the study, we compared the two groups and statistically analyzed the following parameters:Duration of hospitalization;Type of anesthesia;Rate of postoperative complications.

To describe the relationships between qualitative variables, we used Fisher’s and odds-ratio (OR) tests with a CI of 95% or Chi² and Cramer V.

In the case of quantitative variables, we analyzed the normality of the data distribution using the Shapiro–Wilk test, and we interpreted a *p* < 0.05 = the data are not normally distributed, and *p* > 0.05 = the data are normally distributed.

To study the relationships between quantitative variables and binary (dichotomous) qualitative variables, we used the Mann–Whitney test in case of an abnormal distribution and Welch’s two-sample t-test for unequal variances or the t-test for equal variances for data with a normal distribution. In the case of analyzing more than two groups (nominal qualitative variables), we used the Kruskal–Wallis’s test for data that are not normally distributed. We presented the *p*-values generated by these tests as well as the means ± SD of the groups and the difference in the means with a CI of 95%.

To study the relationships between quantitative variables, which were not normally distributed, we used the correlation coefficient Spearman (R) with the associated *p*-value and plotted the relationships as scatter plots on which we added the regression line.

WPS Office 2019 was used for database management. For all subsequent statistical analyzes and graphs, we used R version 4.0.1 (2020-06-06) (R Core Team, 2020).

## 3. Results

After applying the inclusion and exclusion criteria, a sample of 53 patients was obtained.

### 3.1. Description of the Lot According to Demographic Data

Out of a total of 53 patients, 34 (64.15%) were females, and 19 (35.85%) were males. Therefore, the sex ratio is 1.79: 1 (women: men). Among the patients studied, 30 came from urban environments (65.2%) and 16 from rural environments (34.8%). For the remaining 7 patients, the environment of origin was not specified in the observation sheets. The average age of group A is 53.88 ± 12.40 SD, and the median age is 54, with the minimum age being 25 years and the maximum 76 years. Most patients are in the age range of 40–70 years (75.5%), with a peak frequency in the age group 50–60 years (32.1%).

### 3.2. Description of the Sample According to the Pathology Data

#### 3.2.1. Personal Pathological History of Treated Varicose Veins

Out of 53 patients, 4.77% (9 patients) had a previous personal history of surgical treatment for varicose veins of the lower limbs, and one (0.53%) had undergone radiofrequency thermal ablation for the same objective.

#### 3.2.2. Clinical Diagnosis at Hospitalization

Of the total number of patients, 38 patients (71.7%) presented with acute inflammation and superficial thrombophlebitis. This was in contrast with 12 patients (22.6%) who arrived with uncomplicated varicose veins and 3 patients (5.7%) with venous skin ulcers (Figure 1).

#### 3.2.3. The Affected Lower Limb and Saphenous Vein

In our sample, the right lower limb was more frequently affected by varicose veins (59.6% of cases) compared to the left and was affected only in 40.4% of cases; one patient was not included in this calculation due to an unspecified affected lower limb (Figure 2). 

Based on our evaluation, the GSV was affected in 44 patients (meaning 84.6%), and the SSV was affected in 8 patients (meaning 15.4%). No saphenous vein involved was specified in one patient (Figure 3).

#### 3.2.4. CEAP Classification

The arithmetic mean of the clinical score from the CEAP classification was 2.62 ± 0.965 SD, and the median was 2, considering that 34 (64.2%) patients were admitted to the hospital with a clinical score of 2 in the CEAP classification. By taking the clinical stage at hospital admission into consideration, the values ranged from Class 2 (varicose veins) to Class 6 (active venous skin ulcer). (Figure 4).

#### 3.2.5. Correlation of Hospitalization Cost with Different Indicators of Pathology

The hospitalization cost does not have a consistent correlation with the clinical score in the CEAP classification (Spearman’s rank correlation rho: R = 0.186, *p* = 0.183). Practically, the cost of hospitalization does not increase for those in a more advanced stage of the disease studied. Further studies are required in order to count the costs on a broader spectrum of patients.

### 3.3. Description of the Sample According to the Data Related to the Therapeutic Procedure

#### 3.3.1. Type of Anesthesia and Therapeutic Options

Out of the total number of patients (53), 37 (72.5%) used spinal anesthesia, and 14 (27.5%) used general anesthesia. The type of anesthesia used was not specified in two patients.

More than half of the subjects evaluated (27 patients) were treated by the minimally invasive radiofrequency thermal ablation procedure, and 49.1% (26 patients) were treated by the classic, open surgical procedure, which consists of saphenous vein high ligation and stripping. Both types of procedures were accompanied by phlebectomies of the tributaries as a secondary procedure recommended for treating remaining varicose veins.

#### 3.3.2. Duration of Hospitalization

Most of the patients (54.7%) were hospitalized for 2 days. The rest of the patients were hospitalized for a period ranging from 3 to 7 days. The arithmetic mean of the duration of hospitalization was 2.72 days ± 1.03 SD, and the median was 2 days (Figure 5).

#### 3.3.3. Distribution of the Therapeutic Procedure According to Demographic and Pathology Data

According to the applied statistical tests, there is no statistical difference between the group treated by radiofrequency thermal ablation and the one treated surgically in terms of sex (OR = 0.46 [0.15, 1.46] (Fisher test: *p* = 0.254)), age (*t*-test: *p* = 0.202), area of origin (OR = 1.00 [0.30, 3.37] (Fisher test: *p* > 0.999)), CEAP clinical stage at admission (Mann–Whitney test: *p* = 0.485), clinical diagnosis at admission (Cramér V and *p* calculated by Chi^2^ test, V = 0.14 (*p* = 0.586)) and the affected lower limb (OR = 2.25 [0.72, 6.99] (Fisher test: *p* = 0.258)). However, a significant difference was found between the groups relating to the affected saphenous vein (*p* < 0.05). The chance of being treated by RFA for GSV is 10.11 times higher than by high ligation and stripping. For SSV, open surgery is 10.11 times higher than by RFA (OR = 10.11 [1.14, 89.43] (Fisher test: *p* = 0.022)). We can conclude that the two groups are homogeneous in terms of demographic and pathological data, the difference being in the affected saphenous vein. (Figure 6).

#### 3.3.4. Distribution of the Type of Anesthesia According to the Therapeutic Procedure

At first glance, there is a higher use of spinal anesthesia in patients who were open surgically treated than those treated with radiofrequency thermal ablation, but this difference is statistically insignificant (Fisher Test: *p* = 0.127) (Figure 7).

#### 3.3.5. Comparison of the Duration of Hospitalization between the Two Groups

The duration of hospitalization, being a discrete quantitative variable, was shown by the Shapiro–Wilk normality test: *p* < 0.001 and the analyzed data are not normally distributed. Consequently, to analyze whether there is a statistical difference in the length of hospitalization for the two groups of patients, depending on the operating procedure, the Wilcoxon rank sum test with continuity correction was applied. We concluded that the group treated with RFA proved a shorter hospitalization period than the one treated with the classic surgical procedure, with a tendency of statistical significance (0.05 < *p* = 0.066 < 0.1). The number of days until hospital discharge does not correlate statistically significantly with the type of anesthesia used. (Wilcoxon rank sum test with continuity correction: *p* = 0.130) (Table 1).

#### 3.3.6. Comparison of Postoperative Complications between the Two Groups

Out of a total of 29 patients with postoperative complications, 13 were in the RFA group and 16 were in the surgical treatment group. The difference in the total complication rate between the two groups does not prove to be statistically significant (OR = 0.58 [0.19, 1.73] (Fisher test: *p* = 0.412)). When evaluating each individual type of complication, a significant difference was found in the case of hematoma or hemorrhage consecutively; the chance of having hematoma or hemorrhage in the RFA treatment group is 14.28 times lower than in the surgical treatment group (OR = 0.07 [0.00, 1.36] (Fisher test: *p* = 0.023)). Hematoma or hemorrhage occurred as a complication in five of the patients who had undergone surgical treatment and were not encountered in any patient treated with RFA (Table 2).

#### 3.3.7. Duration of Hospitalization Depending on Postoperative Complications

The duration of hospitalization was significantly longer in patients with postoperative complications (Wilcoxon rank sum test with continuity correction: *p* < 0.001). The mean duration of hospitalization of patients with postoperative complications was 3.14 days ± 1.19 SD, and the median was 3 (min:max = 2:7), compared with a mean duration of 2.21 days ± 0.41 SD and a median of 2 (min:max = 2:3) for patients without postoperative complications. There was a significant difference in the length of hospitalization in the groups with cellulitis (Wilcoxon rank sum test with continuity correction: *p* = 0.032), paresthesia (Wilcoxon rank sum test with continuity correction: *p* = 0.010), pain (Wilcoxon rank sum test with continuity correction: *p* < 0.001) and hematoma or hemorrhage (Wilcoxon rank sum test with continuity correction: *p* = 0.005). No significant difference was found between the group when considering the local inflammation (Wilcoxon rank sum test with continuity correction: *p* = 0.224) (Figure 8).

#### 3.3.8. The Cost of Hospitalization Depends on Postoperative Complications

The cost of hospitalization was significantly higher in the group that had postoperative complications (Wilcoxon rank sum exact test: *p* < 0.001). The cost of hospitalization was higher for the groups that had cellulitis (Wilcoxon rank sum exact test: *p* = 0.048), paresthesia (Wilcoxon rank sum test with continuity correction: *p* = 0.013), pain (Wilcoxon rank sum exact test: *p* < 0.001) and hematoma or hemorrhage (Wilcoxon rank sum exact test: *p* = 0.005). There was no significant difference between the groups, based on the existence of postoperative local inflammation (Wilcoxon rank sum exact test: *p* = 0.131) (Figure 9).

## 4. Discussion

Our study highlighted a ratio of women:men of 1.79:1, which, although it cannot be extrapolated to estimate the incidence of varicose veins in the general population, expresses the gender difference relating to this type of vascular pathology [9,10]. In the Edinburgh Vein study, conducted by Robertson et al. in 2013 [11], a prospective cohort study of 1456 patients over a 13-year period, there was no statistically significant difference in the incidence of lower limb varicose veins between male (15.2%) and female patients (17.4%) (*p* = 0.97). On the other hand, in the literature, the female gender is presented as a risk factor for varicose veins, which is consistent with our study, although there is no consensus on the definition of varicose veins [12,13,14].

The average age of the patients included in this study was 53.88, comparable to the average age of 58.2 in a study conducted in 2018 on the Romanian population by Feodor et al. [8]. In a study by Criqui et al. from 2002, over 75% of the patients were over 50 years old [15].

Most patients in the present study came from urban environments (65.2%), which could be due to a more sedentary lifestyle, this being considered a risk factor for varicose veins in the literature [16,17,18].

One study, which was conducted by Michaels et al. in 2006, showed the presence of reflux at the level of the GSV in 85% of the evaluated lower limbs, and 20% of the lower limbs had reflux at the SSV level [19]. These results are consistent with our study, with 84.6% of patients experiencing reflux of the GSV and the remaining 15.4% of the SSV.

In the present study, the majority of patients were admitted to the hospital in C2 of the disease, considering the CEAP classification, that of visible varicose veins (71.7%) [20,21,22].

This is comparable with the meta-analysis published by Salim et al. in 2021 [23], which shows for each C class of the CEAP classification the following percentages: C0S: 9%, C1: 26%, C2: 19%, C3: 8%, C4: 4%, C5: 1% and C6: 0.4%, which is consistent with our study.

The chance of being treated by RFA for GSV is 10.11 times higher than by surgical treatment. There was no significant difference in the type of anesthesia used between the two groups (*p* = 0.127). The European College of Phlebology stated in their guideline for truncal ablation that RFA is the most used endovenous technique [24]. When compared with other treatment options, RFA has a better outcome in terms of a surgical success rate when compared with high ligation and stripping or with laser ablation [25]. This is why RFA has been introduced as the treatment of choice in the European Society for Vascular Surgery (ESVS) 2022 Clinical Practice Guidelines on the management of chronic venous disease of the lower limbs [4] and as an appropriate treatment for truncal reflux of the GSV and SSV by the American Venous Forum, the Society for Vascular Surgery, the American Vein and Lymphatic Society, and the Society of Interventional Radiology [5]. These randomized clinical trials have given conservative treatment surgery a Grade A recommendation in the treatment of patients with varicose veins caused by GSV [4,26].

The total rate of complications after radiofrequency thermal ablation was 48.1%, of which 29.6% was local inflammation, 14.8% cellulitis, 3.7% paresthesia and 3.7% pain. There were no cases of skin burns, hyperpigmentation, hematoma or hemorrhage, deep vein thrombosis or pulmonary thromboembolism. In comparison, Proebstle et al. [27,28] initially reported, after 6 months of the clinical trial, a low complication rate of bruising (6.4%), paresthesia (3.2%), hyperpigmentation (2%), hematoma (1.6%), erythema (1.6%) and phlebitis (0.8%). No skin burns or deep vein thrombosis were observed by Mekako et al. in a study from 2006 [29].

In the case of surgical treatment, the total complication rate in our study was 61.5%, with 34.6% local inflammation, 23.1% cellulitis, 7.7% paresthesia, 11.5% pain and 19.2% hematoma or hemorrhage. A prospective study in 1992, performed on 1300 patients and which evaluated the method of stripping of the GSV by invagination, showed a very low rate of hematoma-type complications (0.46%) and paresthesia-type (0.31%) (Staelens et al., 1992) [30]. Compared to this study, the complication rate in our study was higher.

In a randomized clinical trial conducted in 2010 on a group of 180 patients with venous reflux at the saphenous-femoral junction and GSV on DUS examination, group A, treated by radiofrequency thermal ablation and group B, treated by classic surgical treatment (GSV stripping), were compared. Group A, treated with radiofrequency thermal ablation, had a significantly lower overall complication rate (*p* = 0.02) and a significantly shorter hospital stay (*p* = 0.001). In contrast, group A, when treated with RFA, had a significantly increased cost of hospitalization (*p* = 0.003) (ElKaffas et al., 2010) [31]. In our study, the total complication rate did not significantly differ between the two groups (*p* = 0.412), but hospital stays in the radiofrequency thermal ablation group were indeed shorter (*p* = 0.066). There was no statistically significant difference in the cost of hospitalization between the two groups in our study (*p* = 0.639). A lower rate of postoperative complication, such as hematoma or hemorrhage, was identified in the group treated with radiofrequency thermal ablation with an OR = 0.07 (*p* = 0.023) [28].

### Limitations of the Study

-Missing data in the observation papers, which are used in studies to quantify the evolution and rate of postoperative complications; among these data, we can list: CEAP clinical stage at discharge, venous clinical severity test or Aberdeen score at hospitalization and discharge, pre- and postoperative quality of life assessment scores, pre- and postoperative pain assessment scales, consultation sheets at different intervals of intervention attached to the patient’s file, the occurrence of recurrence or postoperative reintervention, postoperative Duplex ultrasound evaluation to highlight the success rate, duration of the intervention.-Missing data related to some patients, such as the personal physiological history in women to estimate the importance of pregnancy as a risk factor, hereditary history to quantify the importance of genetic factors, and the conclusion of the preoperative DUS evaluation.

One major limitation of this study is the small sample size of the patients. Probably, with a larger number of patients, the results in terms of complications between the two methods will change in favor of RFA. Pain is a very subjective perception and depends on the psychological particularities of each patient. For an objective pain assessment, we need to use some special tools, such as the VAS (Visual Analogic Scale); however, this is a subject of future study. In our study, for example, pain and paresthesia are comparable to other studies listed in the references [27,28,29]. Finally, we have taken into account only the early complications in the immediate postoperative days (days 1 and 2), but as time passes, complications reduce faster in the RFA group compared with the open surgical group.

## 5. Conclusions

In conclusion, radiofrequency thermal ablation is a less invasive and more effective alternative to open surgical techniques for treating truncal veins. This technique utilizes heat energy to seal the veins and provides patients with quicker recovery times, reduced pain and scarring, and improved cosmetic results. Despite its numerous benefits, it is important to note that not all patients are suitable candidates for this procedure, and careful selection and evaluation are necessary to determine the best treatment option. Nevertheless, the use of radiofrequency thermal ablation for truncal veins is a step forward in the evolution of minimally invasive vein treatment options. Although it was introduced more than 15 years ago, in our country, it is still low in use. Because chronic venous insufficiency is complex and prone to worsening, it can substantially impact a patient’s QoL. It is important that patients receive appropriate treatment to prevent disease progression and recurrence. Thus, referral of these patients to a specialist may be warranted. Further, it is important for physicians to consider the options and costs of available treatments.

## Figures and Tables

**Figure 1 ijerph-20-03308-f001:**
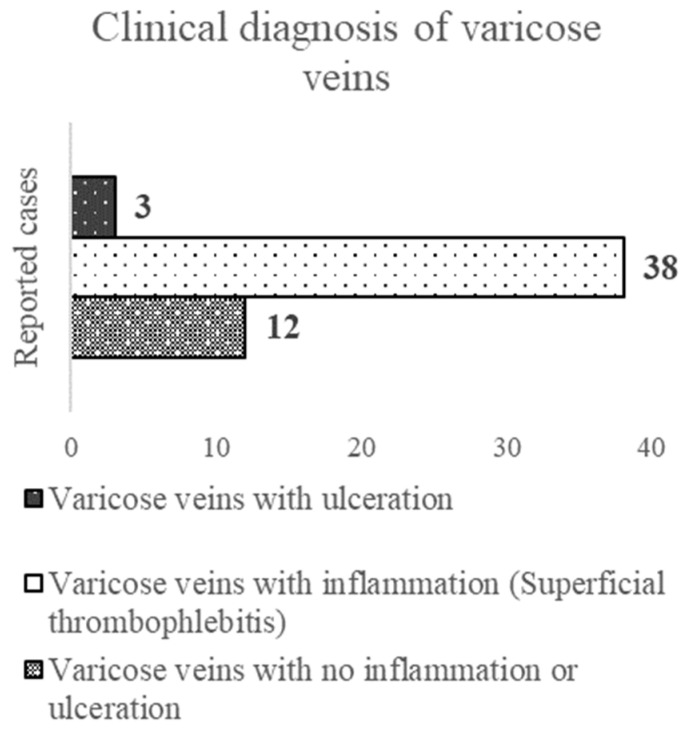
Clinical description of varicose veins.

**Figure 2 ijerph-20-03308-f002:**
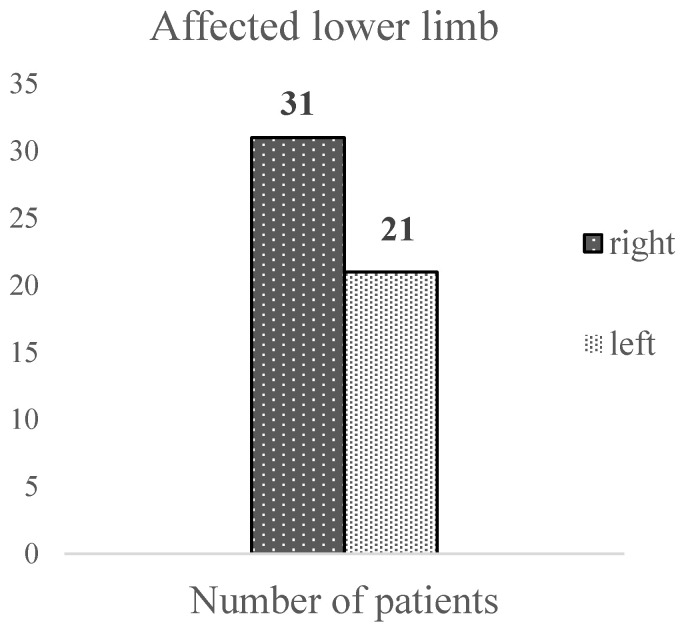
The site of the varicose veins.

**Figure 3 ijerph-20-03308-f003:**
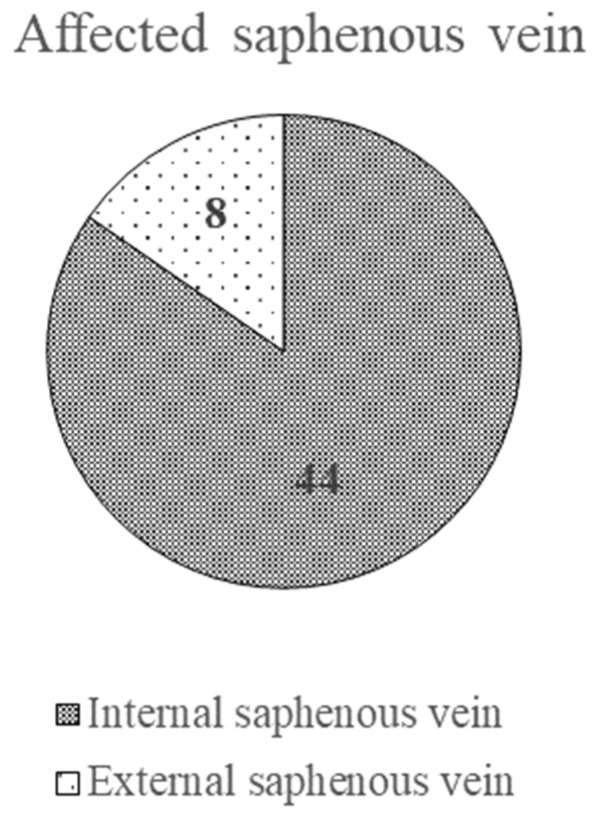
The affected saphenous vein.

**Figure 4 ijerph-20-03308-f004:**
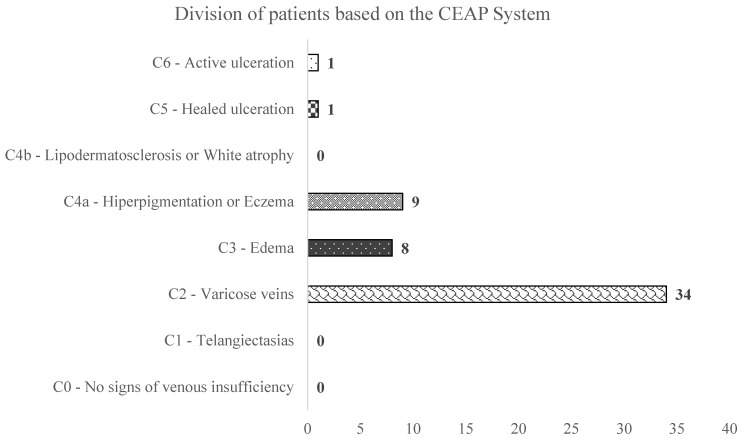
The distribution of patients according to the CEAP classification.

**Figure 5 ijerph-20-03308-f005:**
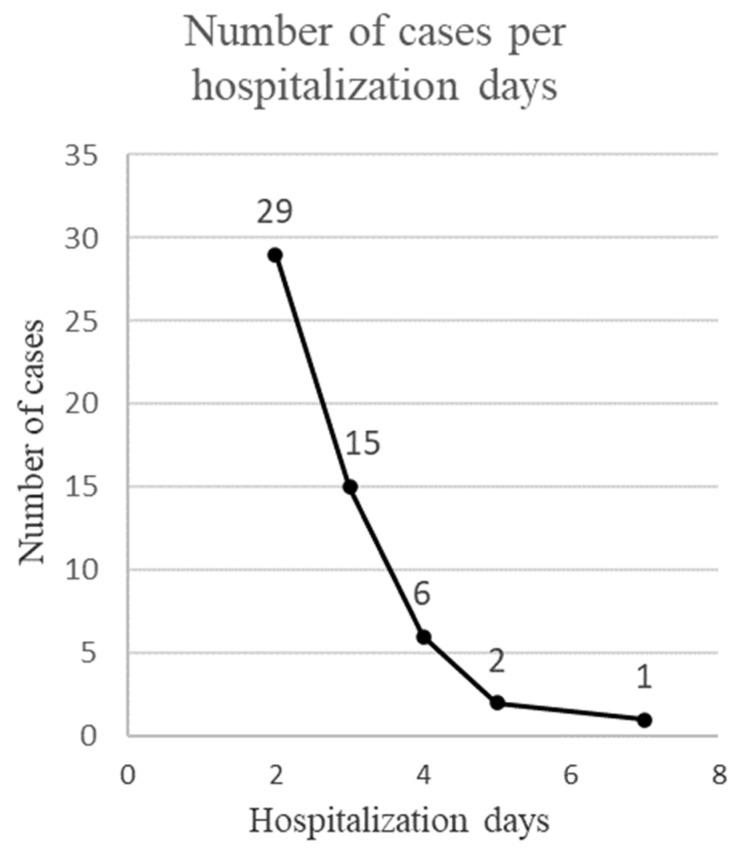
Duration of hospitalization.

**Figure 6 ijerph-20-03308-f006:**
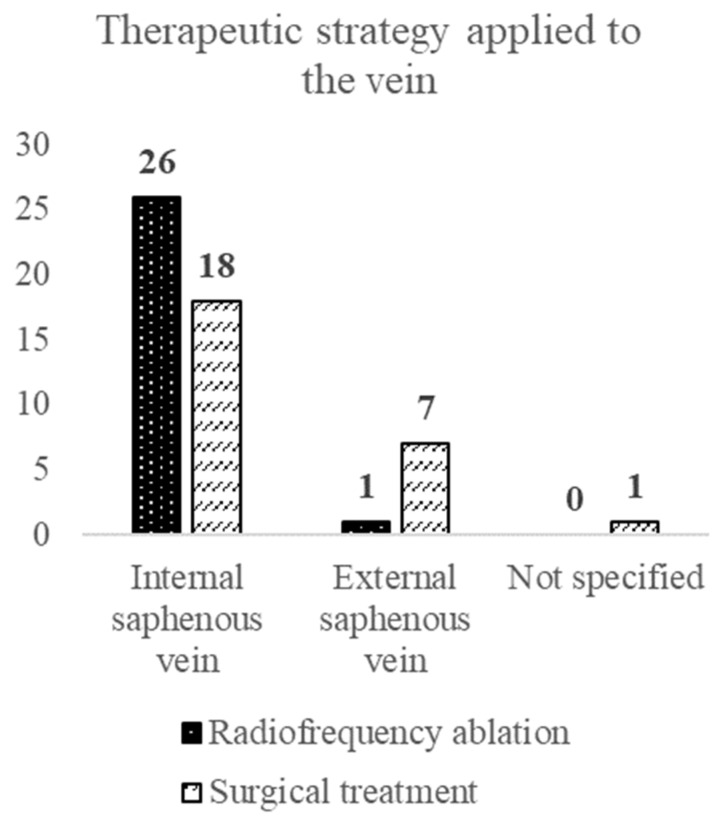
Treatment option and saphenous vein type.

**Figure 7 ijerph-20-03308-f007:**
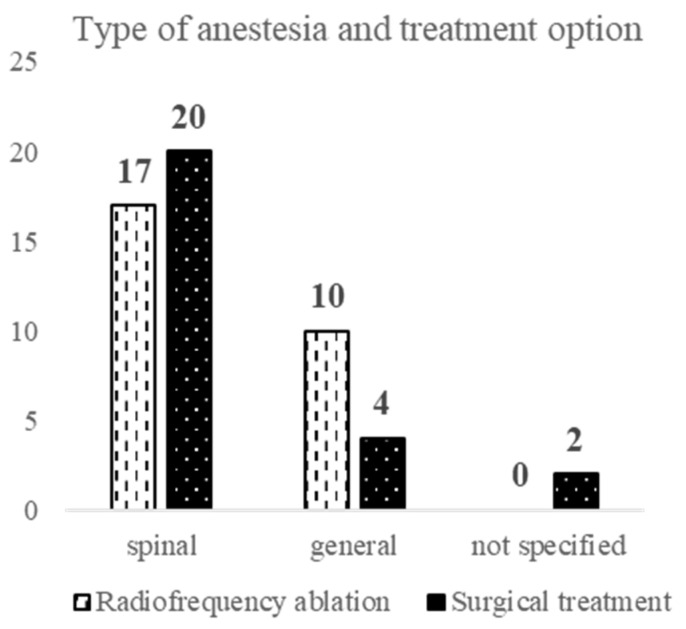
Anesthesia and treatment selection.

**Figure 8 ijerph-20-03308-f008:**
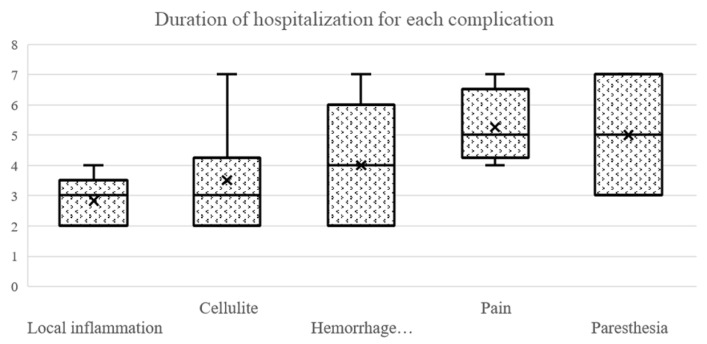
Duration of hospitalization for each complication.

**Figure 9 ijerph-20-03308-f009:**
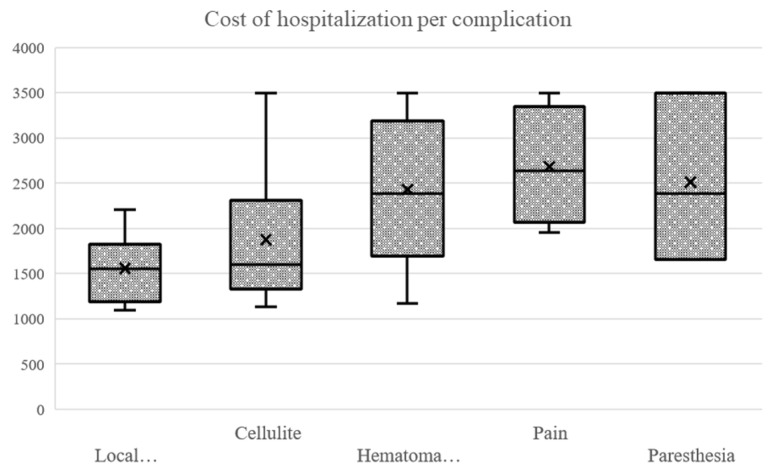
Cost of hospitalization based on postoperative complications.

**Table 1 ijerph-20-03308-t001:** Duration of hospitalization between the two treatment groups.

Subset	Number of Cases	Mean ± SD	Med (Min:Max)
Duration of hospitalization (days) (Shapiro–Wilk normality test: *p* < 0.001)			
Total	53 (100.0%)	2.717 ± 1.03	2.00 (2.00:7.0)
Procedure (Wilcoxon rank sum test with continuity correction: *p* = 0.066)			
Radiofrequency ablation	27 (50.9%)	2.407 ± 0.57	2.00 (2.00:4.0)
Surgical treatment	26 (49.1%)	3.038 ± 1.28	3.00 (2.00:7.0)
Anesthesia (Wilcoxon rank sum test with continuity correction: *p* = 0.130)			
General	14 (27.5%)	2.357 ± 0.63	2.00 (2.00:4.0)
Spinal	37 (72.5%)	2.838 ± 1.12	3.00 (2.00:7.0)

**Table 2 ijerph-20-03308-t002:** Treatment strategy and postoperative complications reported.

	Radiofrequency Ablation	Surgical Treatment	Total	Statistical Analysis
Procedures	27 (50.9%)	26 (49.1%)	53	
Postoperative complications	13 (48.1%)	16 (61.5%)	29 (54.7%)	OR = 0.58 [0.19, 1.73] (*p* = 0.412)
Local inflammation	8 (29.6%)	9 (34.6%)	17 (32.1%)	OR = 0.80 [0.25, 2.53] (*p* = 0.773)
Cellulitis	4 (14.8%)	6 (23.1%)	10 (18.9%)	OR = 0.58 [0.14, 2.35] (*p* = 0.501)
Paresthesia	1 (3.7%)	2 (7.7%)	3 (5.7%)	OR = 0.46 [0.04, 5.42] (*p* = 0.610)
Pain	1 (3.7%)	3 (11.5%)	4 (7.5%)	OR = 0.29 [0.03, 3.04] (*p* = 0.351)
Hematoma or hemorrhage	0	5 (19.2%)	5 (9.4%)	OR = 0.07 [0.00, 1.36] (*p* = 0.023)

OR = odds-ratio [CI 95%] and *p* calculated using Fisher’s test.

## Data Availability

Not applicable.

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
