# Peer review of "Radiofrequency Thermal Ablation for the Treatment of Chronic Insufficiency of the Saphenous Vein—A Comparative Retrospective Study"

_ijerph, 2023, doi:10.3390/ijerph20043308_

Round 1
Reviewer 1 Report
1. The article indicates that the study is a comparative study, but the title does not indicate that it is a comparative study.
2. The article mentions that for GSV, the probability of RFA treatment is 10.11 times that of high ligation and stripping. For SSV, open surgery is 10.11 times more effective than RFA. The 2022 clinical practice guidelines of the European Society of Vascular Surgery list RFA as the first choice for GSV and SSV. So for SSV, is surgery or RFA the first choice?
3. In the article, there is no statistical difference in postoperative complications between RFA and surgical treatment except for hematoma. In the reference studies listed, the postoperative complications of RFA are significantly lower than those of surgery. Why do you get different results and can you explain in detail? for a while?
Author Response
- I consider the study as a comparative one, as it compares two different tehniques, high ligation of saphenous vein, respectively minimal invasive technique. It will be possible to chage the title of the article so that the comparative study will be enclosed.
- the first choice for the treatment of both, GSV and SSV are the minimal invasive techniques (RFA or laser ablation). The 10.11 times higher chance to treat SSV refears to our study, in which the most freqcvent treatment was classical open surgery. For SSV thermal ablation has some limitation because its course along the dorsal calf is sometimes more superficial than the GSV and that predispose to more heat damage to the dermis.
- first of all it was already mentioned that the limitation of the study is the small sample size of the patients. Probably with an increased number the results in terms of complications between the two methods will change in the favour of RFA. Pain is a very subjective perception and depends on the psychological particularities of each patient.For an objective pain assesment we need to use some special tools such as the VAS (Visual Analogic Scale), but this is a subject of a future study. In our study for example pain and paresthesia are comparable to other studies listed in the references (27, 28, 29). Finally we have taken into account only the early complications in the immediate postoperative days (day 1 and 2), but as the time passes, complications reduces more faster in the RFA group compared with the open surgical group
Reviewer 2 Report
The authors must revise the figures. It is noted that in most of the figures, the x axis or y-axis is not visible. Please revise and update with new figures. The radiofrequency procedure technique is not mentioned. Please discuss briefly what is Radiofrequency Ablation? How it can be achieved? What is the clinical definition of thermal ablation? What are the temperature ranges under which the treatment was performed on patients under this study. One thermal ablation study discusses how ablation should be controlled to the restrictive margins [https://doi.org/10.1016/j.cmpb.2020.105781]. Please discuss and compare or contrast the similar in the present context.
Author Response
For radiofrequency ablation of the saphenous veins, we have used the ClosureFast radiofrequency generator (Medtronic) with a 7 Fr 100 cm long ClosureFast catheter. The inserted catheter has an active tip in a lenght of 7 cm which heats a 7 cm vein segment in one 20-second interval to shrink and collapse the target veins. The temperature is kept at a stable 120 degrees Celsius during the 20-second treatment cycle. The catheter delivers the energy which cause collagen contraction, that finally leads to obliteration of the lumen through endothelial destruction, inflammatory response, fibrosis, and permanent occlusion.
The patient is placed in reverse Trendelenburg position in order to get a proper vein distension. Ultrasound-guided access of the great saphenous vein is obtained using a micropuncture needle and .018-inch guide wire at the level of the distal thigh or just below the knee. A 7-Fr sheath is placed over the wire, the ClosureFast™ catheter is then placed through the 7-Fr sheath and advanced to the level of the SFJ. The distance between SFJ and the tip of the catheter should be between 15-20 mm.
The entire treatment length of the vein must be circumferentially injectedwith a normal saline solution prior to begining ablation. The superficial fascia anterior to the vein and the muscular fascia posterior the vein create the saphenous canal. Using ultrasound guidance, the tumescent solution of about 300 to 450 ml of 0.9% saline, is injected circumferentially around the vein in the saphenous canal, which is between the superficial fascia anterior to the vein and the posterior fasscia to the vein, which separates it from the muscle.
RFA may begin after appropriate instillation of tumescent solution along the entire treatment length of the vein and final positioning of the RF catheter is rechecked and confirmed to be 15-20 mm from the saphenofemoral junction (SFJ). Next, the catheter is retracted in 6.5 cm segments, and each vein segment is treated with a single 20 second cycle.
After the final segment of vein has been treated, the catheter and the sheath are removed and manual pressure is applied at the sheath insertion to achieve hemostasis. An elastic stocking or bandage is applied for 24 hours postoperatively.
Regarding the article you mentioned, it describs a different aplication of radiofreqvency, respective in tumour treatment. In venous ablation, damage is limited, as posssible at the venous wall. This can be achieved be injection of tumescent solution around the vein, the role of it consist in absobtion of excessive heat.
The figures will be corrected and updated in the final version of the article